Identification of ABF/AREB gene family in tomato (Solanum lycopersicum L.) and functional analysis of ABF/AREB in response to ABA and abiotic stresses

Pan Xuejuan
Wang Chunlei
Liu Zesheng
Gao Rong
Feng Li
Li Ailing
Yao Kangding
Liao Weibiao liaowb@gsau.edu.cn
College of Horticulture, Gansu Agricultural University , Lanzhou , China
Lal Mohan
Electronic publication date: 2023 May 4
Publication date: 2023
Volume: 11
Electronic Location ID: e15310
Received 2022 Dec 15; Accepted 2023 Apr 6
Copyright: © 2023 Pan et al.
Copyright year: 2023
Copyright holder: Pan et al.
License: This is an open access article distributed under the terms of the Creative Commons Attribution License, which permits unrestricted use, distribution, reproduction and adaptation in any medium and for any purpose provided that it is properly attributed. For attribution, the original author(s), title, publication source (PeerJ) and either DOI or URL of the article must be cited.
License URL: https://creativecommons.org/licenses/by/4.0/

Keywords: ABF/AREB, Abiotic stress, Abscisic acid, Gene family analysis, Tomato, Transcription factor

Funding: National Natural Science Foundation of China 32072559, 31860568, 31560563 and 31160398 Key Research and Development Program of Gansu Province, China 21YF5WA096 National Key Research and Development Program 2018YFD1000800 Research Fund of Higher Education of Gansu, China 2018C-14 and 2019B-082 Natural Science Foundation of Gansu Province, China 1606RJZA073 This work was supported by the National Natural Science Foundation of China (Nos. 32072559, 31860568, 31560563 and 31160398); the Key Research and Development Program of Gansu Province, China (No. 21YF5WA096); the National Key Research and Development Program (2018YFD1000800); the Research Fund of Higher Education of Gansu, China (No. 2018C-14 and 2019B-082); and the Natural Science Foundation of Gansu Province, China (Nos. 1606RJZA073). The funders had no role in study design, data collection and analysis, decision to publish, or preparation of the manuscript.

==============================
Abscisic acid (ABA) is a plant hormone that plays an important regulatory role in plant growth and stress response. The AREB (ABA-responsive element binding protein)/ABF (ABRE-binding factor) are important ABA-signaling components that participate in abiotic stress response. However, genome-scale analysis of ABF/AREB has not been systemically investigated in tomato. This study was conducted to identify tomato ABF/AREB family members and analyze their response to ABA and abiotic stresses. The results show that a total of 10 ABF/AREB members were identified in tomato, which are randomly distributed on five chromosomes. Domain analysis showed that these members exhibit high protein similarity, especially in the basic leucine zipper (bZIP) domain region. Subcellular localization analysis indicated that all 10 ABF/AREB members are localized in the nucleus. Phylogenetic tree analysis showed that tomato ABF/AREB genes are divided into two groups, and they are similar with the orthologs of other plants. The analysis of cis-acting elements showed that most tomato ABF/AREB genes contain a variety of hormones and stress-related elements. Expression profiles of different tissues indicated that SlABF2 and SlABF10 play an important role in fruit ripening. Finally, qRT-PCR analysis revealed that 10 tomato ABF/AREB genes respond to ABA, with SlABF3 being the most sensitive. SlABF3, SlABF5 and SlABF10 positively respond to salt and cold stresses. SlABF1, SlABF3 and SlABF10 are significantly induced under UV radiation treatment. SlABF3 and SlABF5 are significantly induced in osmotic stress. Overall, this study may provide insight into the role of tomato ABF/AREB homologues in plant response to abiotic stresses, which laid a foundation for future functional study of ABF/AREB in tomato.

Introduction

During the whole process of plant growth, there are many abiotic stress factors that hinder plant growth and reduce yield (Zhu, 2016). In different environments, plants produce different stress responses through perception and adaptation. When plants lack water resources, their growth, survival, distribution and productivity will be seriously affected (Fleta-Soriano & Munne-Bosch, 2016). Soil salinization and drought are also important unfavorable factors affecting agricultural development. Moreover, drought can cause damage to the above-ground and underground tissues and organs of plants, as well as change the normal relationship between plant and soil (Qian et al., 2019).

ABA is a plant hormone that is most important in regulating plant adaptation to adversity. By preserving tissue water balance, shutting stomata, promoting root permeability, and enhancing water conductivity, ABA primarily boosts plant resistance (Kudoyarova et al., 2011). ABF (AREB binding factors)/AREBs (ABA response element binding protein) belong to the A subfamily of basic leucine zipper bZIP (Basic leucine zipper) transcription factors. They are key regulatory molecules downstream of the ABA signaling pathway regulating plant response to hormones and stresses, initiating the expression of downstream genes (Chang et al., 2019). ABFs activates the expression of ABA-regulated genes by binding to ABRE homeopathic elements. In the signal transduction of ABA, ABA binds to pyrabactin resistance 1 (PYR1)/PYR1-like (PYL), leading to the inactivation of protein phosphatase 2C (PP2C), destroying the interaction between PP2C and Snf1-related protein kinase2 (SnRK2) and stimulating the activity of SnRk2 to activate AREB (Fujii et al., 2009; Fujita, Yoshida & Yamaguchi-Shinozaki, 2013; Kim et al., 2004). The heterologous overexpression of DcABF3 in carrot increases stomatal density and reduces ABA sensitivity in transgenic Arabidopsis (Wang et al., 2021). In addition, ABA/stress responses are influenced by ABF2, a bZIP protein belonging to the ABF subfamily that interacts with ABA-responsive regions (Kim et al., 2004). It was shown that the overexpression of SlAREB1 exhibits greater tolerance than wild type under high salt stress (Xu et al., 2022). SlAREB may regulate some stress-responsive genes and its overproduction improves plant tolerance to water deficit and salt stress (Hsieh et al., 2010).

Tomato (Solanum lycopersicum L.) has long been used as a model plant for fruit ripening, disease response, genetics and whole genome sequence studies (Hsieh et al., 2010). Around one-third of the world tomato production is in China (Li et al., 2022). Moreover, the most valuable horticultural crop in the world is tomato (Bastias et al., 2011). The production, productivity and quality of tomato are adversely affected by abiotic stress (Krishna et al., 2019). ABF/AREB has a significant function in plant development, growth, and resistance to biotic and abiotic stressors, and this role is gradually being investigated. So far, the existence of ABF/AREB has been identified in many crop plants, including Arabidopsis, potato (Liu et al., 2019), rice (Lu et al., 2009), cotton (Kerr et al., 2018), apple (Ma et al., 2017), and grape (Zandkarimi et al., 2015). However, the information on the members of tomato ABF/AREB subfamily members remains scarce. In order to better understand the key role of the ABF/AREB subfamily in plants, the coding genes of the ABF/AREB members in tomato were identified and analyzed in this study. For tomato ABF/AREB members, the secondary structure, chromosome position of the gene, gene structure, conservative motif analysis, cis-acting elements analysis, phylogenetic relationship and subcellular location analysis were conducted. At the same time, the expression patterns of these ABF/AREBs in different tissue-specific processes and gene transcription analysis under different abiotic stresses and hormones were also investigated. The genome-scale analysis of tomato ABF/AREB is not yet known. Our study here aims to provide a solid basis for help broaden the molecular biological functions of ABF/AREBs in plants, and the potential function of ABF/AREB in growth regulation and abiotic stress alleviation was proposed.

Materials and Methods

Identification of the ABF/AREB family members in tomato

Firstly, the nine Arabidopsis thaliana ABF/AREB (AtABF1, AtABF2/AREB1, AtABF3, AtABF4/AREB2, AtDPBF1/ABI5, AtDPBF2, AtAREB3/DPBF3, AtDPBF4/EEL, and AtbZIP15) amino acid protein conserved sequences, downloaded from TAIR (https://www.arabidopsis.org/), were used to query the ABF/AREB subfamily members of tomato (Li et al., 2020). The members of ABF/AREB gene family were screened as candidate genes by homology comparison in the database of tomato gene testing (https://solgenomics.net/organism/Solanum_lycopersicum/genome). Secondly, an E-value of 1e-20 was used to reduce the false positive, and the PFAM database (http://pfam.xfam.org/) and SMART database (http://smart.embl-heidelberg.de/) were used to further verify the ABF/AREB protein domain of tomato. Candidate genes that did not contain a specific domain of the ABF/AREB gene (PF00170) were manually eliminated. The ABF/AREB gene family was represented by the remaining genes (Liu et al., 2019).

Tomato genome sequence and annotation information were downloaded by Ensemble plants-tomato genome database (http://plants.ensembl.org/index.html). The whole genome information (GFF3, FASTA, PEP, CDS) of tomato was sorted out by using TBtools (toolbox for ecological battle) v1.0985 software (Qu et al., 2021), and finally the whole genome information file of tomato ABF/AREB was screened out for mapping according to the gene ID of the identified members of tomato ABF/AREB gene family.

Characterization of ABF/AREB transcription factor in tomato

The chromosome position, amino acid length, molecular weight, isoelectric point, molecular formula, and other physical and chemical characteristics data were used to examine the tomato ABF/AREB protein sequence (https://web.expasy.org/protparam/). Online analysis of the tomato ABF/AREB transcription factor subcellular location prediction was done using WoLFPSORT (https://wolfpsort.hgc.jp/). The secondary structure of tomato ABF/AREB family proteins were examined using the online website prabi (http://www.prabi.fr/) and the associated data was exported and imaged (Ataee et al., 2022).

Conserved motifs and protein conserved domain analysis

MEME (http://meme-suite.org/tools/meme) was used to input all protein sequences of tomato ABF/AREB online program to analyze tomato ABF/AREB transcription factor family conserved motifs (Chen et al., 2018). The number of predicted motifs was set to 10, while the other parameters were set as default. The multiple sequence alignment of tomato family was done through ClustalX and GeneDox software.

Phylogenetic tree and cis-acting elements analysis

ABF/AREB protein sequences of Arabidopsis thaliana, Solanum tuberosum, and Populus orientalis were obtained from TAIR (https://www.arabidopsis.org/). Plant Gene Database (https://phytozome.jgi.doe.gov) (Liu et al., 2019) and article (Yong et al., 2021) (File S1). Phylogenetic trees were constructed using Mega 7.0 software. A phylogenetic tree of 33 ABF/AREB protein sequences was constructed by neighbor-joining method (Bootstrap parameter was set to 1,000) (Rusinko & McPartlon, 2017). In addition, the evolutionary tree was beautified using EvolView (https://evolgenius.info//evolview-v2/#login) website. A DNA sequence of 2,000 bp upstream of the tomato ABF/AREB gene was obtained from the genome-wide information of tomato ABF/AREB and submitted to PlantCare online database (http://bioinformatics.psb.ugent.be/) for analysis. The cis-acting element analysis results of PlantCARE website were deleted and integrated, and TBtools software was used for analysis (Zhao et al., 2023).

Tissue expression analysis of ABF/AREB gene in tomato

The IDs of the SlABF genes were searched in the eFP (http://bar.utoronto.ca/efp/cgi-bin/efpWeb.cgi) database. Then, the data were sorted out and the expression patterns of SlABF in different tissues were drawn by TBtools (Kim et al., 2013).

Gene location, Ka (nonsynonymous)/Ks (synonymous) analysis and gene structure analysis

The GFF3 file in the whole genome information of tomato ABF/AREB was visualized and analyzed by TBtools software. ABF/AREB gene members were renamed according to their chromosomal distribution (Duan et al., 2022). The CDS sequences of tomato ABF/AREB genes were further used to calculate the relationship between Ka (non-synonymous substitution rate) and Ks (synonymous substitution rate) among family members by calculating Ka and Ks values by Simple Ka/Ks calculator (NG) of TBtools software (Liu et al., 2022b). The exon-intron structure distribution of ABF/AREB gene in tomato was analyzed by using the GFF3 file of ABF/AREB genome-wide information with TBtools software (Chen et al., 2020).

Transcriptional analysis of ABF/AREB gene in tomato under different abiotic stresses and hormone treatments

Plant materials and treatment

Tomato (Lycopersicum esculentum L. ‘Micro-Tom’) seeds were provided by the Institute of Vegetable Science, College of Horticulture, Gansu Agricultural University (about 1,530 m; 36.10384 N, 103.7189 E). The seeds were put in a 250 mL Erlenmeyer flask with 100 mL of sterile water (45 °C), soaked for 10 min, and then put into a high-temperature shake flask at a speed of 180 r min−1 (25 °C). The sterile water was changed one time every day. After germination, the tomato seeds were transferred to a plug tray containing the substrate. After the cotyledons were fully unfolded, the nutrient solution was irrigated every 2 days. The control growth chamber environment had a photoperiod of 16/8 h (light/dark), an air temperature of 26/20 °C (day/night), and a light intensity of 250 µ mol m−2 S−1. Stress treatments were carried out at the two-leaf stage of seedlings. Seedlings were transplanted into solution containing NaCl (200 M), ABA (100 M), fluridone (an inhibitor of ABA biosynthesis) (100 M) and PEG 6000 (20%) for 0, 12 and 24 h. For cold treatment, the seedlings were placed in 1/2 Hoagland nutrient solution and placed in a refrigerator (Qingdao Haier Special Electric Appliance Co., Ltd., Qingdao, China) at 4 °C for 0, 12 and 24 h. Seedlings at the dicotyledonous stage were transferred to the 253.7 nm UV treatment and other growth conditions were the same as those of the control. Leaf samples were collected for qRT-PCR experiments after 0, 12, and 24 h of treatment, and leaves under NaCl, ABA, FLD, PEG, cold and UV treatments were collected (Liu et al., 2022a). The collected samples were immediately frozen in liquid nitrogen and stored in −80 °C vertical ultra-low temperature refrigerator (Qingdao Haier Special Electric Appliance Co., Ltd., Qingdao, China). Each treatment contained three biological replicates.

RNA extraction and quantitative qRT-PCR

Total RNA was extracted from the samples using TRIzol reagent (Invitrogen, Carlsbad, CA, USA) Take advantage of FastQuant First Strand cDNA Synthesis Kit (Tianen, Beijing, China) to synthesize cDNA. These reactions were executed under the following conditions: 37 °C for 15 min, 85 °C for 5 s, and finally ended at 4 °C. LightCycler 480 Real-Time PCR System (Roche Applied Science, Penzberg, Germany) and SYBR Green Premix Pro Taq HS Premix kit was used for qRT-PCR. The reaction system was 2×SYBR Green Pro Taq HS Premix 10 μL, primer F 0.4 μL, primer R 0.4 μL, cDNA 2 μL, ddH2O 7.2 μL. The primers used in qRT-PCR were designed with Primer 5.0, and the internal reference was Actin (NC 015447.3) as shown in Table 1.

Table 1 qRT-PCR primers for expression analysis of ABF/AREB gene family in tomato.

Gene	Prime sequence	Size/bp	
SlABF1	F: ACTACTTGGTGAAAGCCGGG	R: CGATGTCCATAGCACCCCTC	171	
SlABF2	F: GCTACACAGCAGAAACAGCG	R: CCATGATCTGCTTAAGTCTCTCCT	181	
SlABF3	F: CACATTGACATGTCGTGCGAA	R: GTTGCCTTGCAGCTCTGATG	171	
SlABF4	F: TTGGAGGCGACTTCCATGAC	R: ATCCACCGTCCTCCTAACCA	189	
SlABF5	F: GTTTAGGAGCCAGTGGGGTC	R: CTGCCTCCTTTCAACGACCT	176	
SlABF6	F: CAGCAACAGAACAACGGGTG	R:TGATTGCTGCTGAGGAGGTG	162	
SlABF7	F: CAGCAACCAACTCAAAGCCC	R: GCCAGTTGGCAATTGTTCCC	176	
SlABF8	F: GAAAGGAGGCAGAAGCGGAT	R: GCTCTGGAGGTGGAACACTC	178	
SlABF9	F: TGTTGGGCACATTATCGGACA	R: CGAGGCGTGAAACCTTGTTC	183	
SlABF10	F: GCGTTGTCATCTTCTGCTGC	R: CTCCCAAGGTAGATTCCCGC	187	
Actin	F: AATGAACTTCGTGTGGCTCCAGAG	R: ATGGCAGGGGTGTTGAAGGTTTC		

Data statistics and analysis

The data were analyzed using the 2−∆∆Ct calculation method. GraphPad Prism software was used for statistical analysis (Schmittgen & Livak, 2008). ANOVA was used to detect the significant level of difference between different times or different treatments (Bertinetto, Engel & Jansen, 2020) (File S2).

Results

Identifification of ABF/AREB genes in tomato

Ten tomatoes ABF/AREB genes were obtained by homologous alignment, which were named SlABF1-SlABF10 according to the location of the genes on the different chromosomes (Table 2). The tomato ABF/AREB transcription factor family is unevenly distributed on five chromosomes of tomato. Among them, SlABF1, SlABF2 and SlABF3 are located on Chr-01, SlABF4 and SlABF5 are located on Chr-04, SlABF6 is located on Chr-09. SlABF7, SlABF8 and SlABF9 are distributed on Chr-10 and SlABF10 is located on Chr-11 (Fig. 1). To investigate the selection pressure during the evolution of ABF/AREB genes, we calculated Ka/Ks values of tomato ABF/AREB (File S3). The results showed that three pairs of replication genes (SlABF7/SlABF9, SlABF7/SlABF10 and SlABF8/SlABF9) in the tomato ABF/AREB family had Ka/Ks ratios less than one (between 0.24 and 0.30), with two pairs of tandem repeats and one pair of fragment repeats. This indicates that the tomato ABF/AREB gene family underwent purifying selection after the replication event. A Ka/Ks value of less than one implies purifying selection, Ka/Ks = 1 represents neutral selection and Ka/Ks >1 indicates positive selection (Yong et al., 2021).

Table 2 Information of the ABF/AREB transcription factors in tomato.

Gene	Gene ID	Gene locus	ORF
(bp)	Amino acid	Instability
index	Molecular
weight/kDa	pI	Subcellular
localization	
SlABF1	Solyc01g008980.3.1.ITAG3.2	Chr01	441	146	55.95	16,689.92	9.22	Nucleus	
SlABF2	Solyc01g104650.3.1.ITAG3.2	Chr01	894	297	66.49	32,080.06	7.81	Nucleus chloroplast	
SlABF3	Solyc01g108080.3.1.ITAG3.2	Chr01	1,245	414	58.79	45,028.31	9.64	Nucleus	
SlABF4	Solyc04g071510.3.1.ITAG3.2	Chr04	927	308	56.19	33,874.15	6.71	Nucleus	
SlABF5	Solyc04g078840.3.1.ITAG3.2	Chr04	1,344	447	50.23	47,977.73	9.42	Nucleus	
SlABF6	Solyc09g009490.3.1.ITAG3.2	Chr09	1,281	426	54.70	46,072.80	8.79	Nucleus	
SlABF7	Solyc10g050210.2.1.ITAG3.2	Chr10	1,137	378	52.09	41,170.31	9.72	Nucleus	
SlABF8	Solyc10g076920.2.1.ITAG3.2	Chr10	975	324	62.54	36,282.85	6.41	Nucleus	
SlABF9	Solyc10g081350.2.1.ITAG3.2	Chr10	1,053	350	55.95	38,396.10	8.63	Nucleus	
SlABF10	Solyc11g044560.2.1.ITAG3.2	Chr11	1,098	365	61.45	40,007.81	8.51	Nucleus	

Figure 1 The distribution of ABF/AREB gene family members of chromosomes in tomato.

Chromosome positioning was based on the physical location of the 10 tomato ABF/AREBs. Chromosome numbers are shown at the top of each bar chart. Gene names are indicated in black. The scale bar is on the left.

The amino acid size of tomato ABF/AREB transcription factor family is between 146 aa (SlABF1) and 447 aa (SlABF5). The molecular weight is between 16,689.92 and 47,977.73 kDa. The isoelectric point (pI) is between 6.41 (SlABF5) and 9.72 (SlABF4). In the ABF/AREB family, only SlABF4 and SlABF8 are acidic proteins (pI < 7), and the rest are alkalescent (pI > 7) (Table 2). The instability index is greater than 40 for all 10 tomato genes, showing that the ABF/AREB genes are unstable proteins. From the perspective of subcellular location analysis, SlABF1-S1ABF10 are all expressed in the nucleus, which speculate that the gene is related to the storage and replication of genetic material. SlABF2 is expressed in chloroplasts, suggesting that SlABF2 may be involved in photosynthesis.

Conserved domain and conserved motifs of tomato ABF/AREB family

ABF/AREB has a highly conserved protein structure including four conserved phosphorylation sites, three conserved domains of C1, C2 and C3 at the N-terminus, and a highly conserved domain of C4 at the C-terminus (Fujita et al., 2005). The tomato ABF/AREB protein has four conserved phosphorylation sites. The N-terminal is made up of C1, C2, and C3, whereas the C-terminal is made up of C4 and the bZIP region (basic region and leucine zipper). The C-terminus of tomato ABF/AREB proteins has the unique BRLZ domain of bZIP transcription factor, which has the function of recognizing and binding specific DNA sequence (Fig. 2).

Figure 2 Multiple sequence alignment of tomato ABF/AREB members.

Residues are shaded in black and light black, respectively. The positions of C1 to C4 are conserved domains and basic regions are represented by lines above the protein sequence. Potential phosphorylated residues (R-S-S-X/T) of the characteristic phosphorylation sites are indicated by red boxes. Positions of conserved Leu residues in Leu zippers. Domains are represented by red triangle.

In this study, 10 conserved motifs are found in the tomato ABF/AREB proteins (Fig. 3). Sequence information for the identified conserved motifs is presented in Table 3, and the amino acid sequences of different conserved motifs are shown by the stack of letters at each position (File S4). The length of each motif is between 10 and 50 amino acids. The results show that the 10 identified tomato ABF/AREB motifs are quite similar. Motif1 is the basal core of the bZIP domain. Motif5 and Motif4 constitute the C1 conserved phosphate site. Motif3, Motif2, and Motif6 constitute the C2, C3, and C4 conserved phosphate sites, respectively. Both Motif1 and Motif2 are presented in all tomato ABF/AREB proteins. Both Motif3 and Motif4 are presented in nine tomato ABF/AREB proteins except for SlABF1. Except for SlABF1 and SlABF10, the other eight tomato ABF/AREB proteins contain Motif6. Motif7 and Motif9 are in four tomato ABF/AREB proteins (SlABF3, SlABF5, SlABF7 and SlABF10). Motif10 occurs in SlABF3, SlABF5 and SlABF7. Motif8 occurs in SlABF8 and SlABF9. It can be inferred that the tomato ABF/AREB members are highly conservative and may have similar functions.

Figure 3 Sequence analysis of ABF/AREB gene family in tomato.

The different colored rectangles are different motifs.

Table 3 Details of the 10 conserved motifs of tomato ABF/AREB proteins.

Motif	Width
(aa)	Motif sequence	
Motif 1	50	EKVVERRQRRMIKNRESAARSRARKQAYTVELEAEVAKLEEENERLKKKK	
Motif 2	26	GZRQSTLGEMTLEDFLVKAGVVREDA	
Motif 3	29	SLQRQGSLTLPRTLSQKTVDEVWRDIQKE	
Motif 4	21	GGLGKDFGSMNMDELLKNIWT	
Motif 5	18	LARQSSIYSLTFDELQNT	
Motif 6	21	LPNVPKREPLRCLRRTLSGPW	
Motif 7	25	NLDTSSLSPSPYAFNEGGRGRKSCS	
Motif 8	50	WSQYQIPAMQPLPPQQHQQQQQNIPPVFMPGHPIQQPLPIVANPIIDAAY	
Motif 9	33	QQQPLFPKQTTVEFASPMQLGNNGQLASPRTRA	
Motif 10	10	MGSYLNFKNF	

Phylogenetic analyses of the tomato ABF/AREB families

The 33 ABF/AREB (10 SlABFs, 9 AtABFs, 7 StABFs and 7 PdABFs) proteins are divided into two subfamilies (Group A and Group B) (Fig. 4). Among them, SlABF3, SlABF5, SlABF7 and SlABF10 belong to Group A. They have the highest homology with StAREB1, StAREB2, StAREB3 and StAREB4, respectively. SlABF1, SlABF2, SlABF4, SlABF6, SlABF8 and SlABF9 belong to Group B. SlABF6 is more closely related to StABI5. SlABF8 is more closely related to StABL2, and SlABF9 is more closely related to StABL1. SlABF2 and SlABF4 are closely related to AtDPBF2. It can be concluded from the entire evolutionary tree that SlABFs have the highest homology with StABFs, relatively low homology with AtABFs, and the lowest homology with PdABFs.

Figure 4 Phylogenetic relationship of tomato ABF/AREB homologs in different species.

Tomatoes are marked as a yellow five-pointed star.

Analysis of the gene structure of tomato ABF/AREB family

By analyzing the phylogenetic tree and gene structure of the tomato ABF/AREB gene family, the overall number of introns and exons of the 10 tomato ABF/AREB genes had no significant difference, all ranging from 1 to 4. Tomato ABF/AREB genes are divided into two groups. SlABF3, SlABF5, SlABF6, SlABF7 and SlABF10 are divided into ClASS I, and SlABF1, SlABF2, SlABF4, SlABF8 and SlABF9 are divided into ClASS II (Fig. 5). We found that the number of exons in tomato ABF/AREB is between two and four. The number of introns is between one and four. Specifically, in ClASS I, SlABF5, SlABF6, and SlABF10 contain four introns and four exons, SlABF7 possesses three introns and three exons, SlABF3 contains one intron and two exons. Both SlABF2 and SlABF4 in ClASS II have three introns and four exons. There are three exons in SlABF1, SlABF8 and SlABF9, and two introns in SlABF1 and SlABF9. SlABF8 has four introns. In general, the genetic structures of different tomato ABF/AREB members are relatively similar. Interestingly, except for SlABF1 and SlABF3, the other eight genes have similar exon lengths. Thus, the function of the tomato ABF/AREB genes may also be relatively similar.

Figure 5 Exon-intron structure of ABF/AREB gene family in tomato.

(A) A phylogenetic tree was constructed based on the full-length tomato ABF/AREB protein sequence using MEGA7.0 software. (B) The exon-intron map of the tomato ABF/AREB gene was drawn using TBtools. Green rectangles represent exons, and orange rectangles represent upstream and downstream noncoding regions of genes. Solid black lines represent introns. The scale bar represents the length of the DNA sequence.

Analysis of protein secondary structure of tomato ABF/AREB family genes

The most abundant protein secondary structures within tomato ABF/AREB members are mainly alpha helix and random coil (Table 4). The 10 ABF/AREB encoded proteins have alpha helix (29.31–50.68%), extended strand (4.79–13.49%), beta turn (1.01–3.14%), and random coil (43.15–59.26%) as their secondary protein structures (File S5).

Table 4 The secondary structure of ABF/AREB gene family protein sequence in tomato.

Blue indicates alpha helix; green indicates beta turn; red indicates extended strand; purple indicates random coil.

Protein	Alpha helix (%)	Extended
strand (%)	Beta turn (%)	Random coil (%)	Distribution of secondary structure elements	
SlABF1	50.68	4.79	1.37	43.15		
SlABF2	40.07	7.07	1.01	51.85		
SlABF3	31.40	11.59	3.14	53.86		
SlABF4	44.48	4.87	1.62	49.03		
SlABF5	29.31	10.07	1.57	59.06		
SlABF6	34.51	6.81	1.41	57.28		
SlABF7	32.54	13.49	2.91	51.06		
SlABF8	33.95	5.25	1.54	59.26		
SlABF9	34.29	7.14	1.43	57.14		
SlABF10	32.88	9.04	1.92	56.16		

Analysis of cis-acting elements of tomato ABF/AREB family genes

The tomato ABF/AREB genes include a total of 18 homeopathic components (Fig. 6 and File S6). Among them, three elements (AE-box, GATA-motif, MRE) are related to light response, seven elements (ABRE, CGTCA-motif, GARE-motif, P-box, TGACG-motif, TCA-element, TATC-box) are related to hormone response, and four elements (ARE, LTR, MBS, TC-rich repeats) are related to stress response (Table 5). In order to further study cis-elements in the ABF/AREB promoter sequences, three main types of cis-acting elements are identified, including light, hormones, and stress response elements (Fig. 7). AE-box element is mainly distributed in SlABF9. ARE element is mainly distributed in SlABF8. ABRE is all tomato ABF/AREBgenes, with the exception of SlABF7 and SlABF8, and was prevalent in SlABF10. Both CGTCA-motif and TGACG-motif elements are mainly distributed in SlABF3 and SlABF7. Generally speaking, the cis-elements correlated to hormone is relatively more abundant, which manifesting that the tomato ABF/AREBs gene plays a vital role in regulating hormone response.

Figure 6 The distribution of cis-acting elements in tomato ABF/AREB genes.

Different colored wedges represent different cis elements. The length and position of each SlABF genes were mapped to scale. The scale bar represents the length of the DNA sequence.

Table 5 Each of the tomato ABF/ABRE gene family has the original function of cis-acting elements.

Cis-element	Number of genes	Sequence of Cis-element	Functions of cis-elements	
ABRE	18	TACGTGTC	Cis-acting element involved in the abscisic acid responsiveness	
ACE	4	CTAACGTATT	Cis-acting element involved in light responsiveness	
AE-box	5	AGAAACAA	Part of a module for light response	
ARE	15	AAACCA	Cis-acting regulatory element essential for the anaerobic induction	
Box 4	32	ATTAAT	Part of a conserved DNA module involved in light responsiveness	
CGTCA-motif	7	CGTCA	Cis-acting regulatory element involved in the MeJA-responsiveness	
GARE-motif	6	TCTGTTG	Gibberellin-responsive element	
GATA-motif	6	GATAGGA	Part of a light responsive element	
G-Box	17	TACGTG	Cis-acting regulatory element involved in light responsiveness	
LTR	5	CCGAAA	Cis-acting element involved in low-temperature responsiveness	
MBS	6	CAACTG	MYB binding site involved in drought-inducibility	
MRE	3	AACCTAA	MYB binding site involved in light responsiveness	
P-box	4	CCTTTTG	Gibberellin-responsive element	
TATC-box	5	TATCCCA	Cis-acting element involved in gibberellin-responsiveness	
TCA-element	5	CCATCTTTTT	Cis-acting element involved in salicylic acid responsiveness	
TC-rich repeats	3	ATTCTCTAAC	Cis-acting element involved in defense and stress responsiveness	
TGACG-motif	7	TGACG	Cis-acting regulatory element involved in the MeJA-responsiveness	
Circadian	5	CAAAGATATC	Cis-acting regulatory element involved in circadian control	

Figure 7 The number of cis-acting elements in tomato ABF/AREB genes.

Tissue-specific expression pattern of tomato ABF/AREB genes

In order to investigate the expression of the ABF/AREB gene in various tomato tissues during various growth stages. The expression of ABF/AREB genes in 14 tomato tissues is analyzed, including unopened flower bud, fully opened flower, leaf, root, 1 cm fruit, 2 cm fruit, 3 cm fruit, mature green fruit, breaker fruit, breaker fruit + 10, pimpinellifolium immature, green fruit, pimpinellifolium breaker fruit, pimpinellifolium breaker + 5 fruit and pimpinellifolium leaf (Fig. 8). Some SlABFs, including SlABF2, SlABF3, and SlABF10, are highly expressed in all tissues. In contrast, SlABF6 and SlABF7 are expressed at low levels in all tissues. The expression level of SlABF5 in roots is much higher than that in other tissues. The expression of SlABF9 is higher in pimpinellifolium leaf, but lower in other tissues. In addition, SlABF1, SlABF4 and SlABF8 also show similar expression patterns.

Figure 8 Expression patterns of tomato ABF/AREB in different tissues.

Color scale represents fold change normalized by log2 transformed data. Heatmaps are shown in blue/yellow/red for low/medium/high expression respectively.

Expression profles analysis ABF/AREB genes in tomato under ABA and FLD treatment

The relative expression of SlABF1, SlABF2, SlABF3, SlABF4, SlABF5, SlABF8, SlABF9 and SlABF10 is significantly up-regulated under ABA and FLD treatments (Fig. 9). SlABF6 expression decreases after 12 h of ABA treatment and then increases gradually. There is a downward trend under the treatment of FLD. SlABF7 is up-regulated by ABA treatment, but increases first and then decreases under FLD treatment. Seven genes (SlABF1, SlABF2, SlABF5, SlABF6, SlABF7, SlABF9, and SlABF10) have higher relative expression levels under ABA than under FLD treatment. When compared to FLD treatment, the relative expression of SlABF3 and SlABF8 under ABA treatment at 12 h is marginally greater, whereas at 24 h, it was marginally lower. At 12 h, the relative expression of SlABF4 is higher in FLD treatment than in ABA treatment, while, at 24 h, it is lower in FLD treatment than in ABA treatment.

Figure 9 Relative expression analysis of SlABF gene under ABA and FLD treatments.

(A) SlABF1; (B) SlABF2; (C) SlABF3; (D) SlABF4; (E) SlABF5; (F) SlABF6; (G) SlABF7; (H) SlABF8; (I) SlABF9; (J) SlABF10. The asterisk (*) indicates that the expression level of the stress group is significantly different from that of the control group (*p < 0.05, **p < 0.01, one-way ANOVA, Tukey test).

Expression profles analysis of ABF/AREB genes in tomato under NaCl, UV, cold and PEG treatments

In order to clarify the role of ABF/AREB in tomato under abiotic stress, the expression levels of 10 ABF/AREB genes in tomato under NaCl, UV, cold and PEG treatments were studied. As shown in Fig. 10A, the relative expression levels of 10 ABF/AREB genes in tomato are different under NaCl treatment. SlABF6 and SlABF7 expression levels is decreased by NaCl and cold treatments (Figs. 10A and 10C). In contrast, the expression of SlABF3, SlABF5 and SlABF10 is upregulated by NaCl and cold treatments. Moreover, SlABF8 is also significantly upregulated by cold stress. Under NaCl treatment, four genes (SlABF1, SlABF2, SlABF3, and SlABF4) reach the highest levels at 12 h, with SlABF3 showing the greatest change and increasing approximately 9.06-fold compared to 0 h. The expression levels of SlABF6, SlABF7 and SlABF9 decrease gradually with the increase of treatment time. The expression levels of the remaining three genes (SlABF5, SlABF8, and SlABF10) are highest at 24 h with NaCl treatment. Under cold treatment, the expression of the four genes (SlABF1, SlABF3, SlABF4 and SlABF5) gradually increases and reaches the highest level at 24 h. Compared to 0 h, it is increased by 2.26, 16.80, 3.59 and 10.56-folds, respectively.

Figure 10 Analysis of relative expression of ABF/AREB genes in tomato under abiotic stresses including NaCl (a), Uv (b), Cold (c) and PEG (d).

The asterisk (*) indicates that the expression level of the stress group is significantly different from that of the control group (*p < 0.05, **p < 0.01, one-way ANOVA, Tukey test).

The relative expression levels of SlABF6, SlABF7 and SlABF9 are significantly inhibited by UV treatment (Fig. 10B). After 12 h, the relative expression levels of SlABF2 and SlABF8 remain essentially unaltered. As the amount of time spent receiving UV treatment is extended, the relative expression levels of SlABF3, SlABF4 and SlABF5 steadily increase. SlABF3 has the largest change trend, increasing by over 38.50 times at 24 h compared to 0 h. After 12 h, SlABF1 and SlABF10 expression levels dramatically increase, and at 24 h, they marginally reduce.

Under PEG treatment, the relative expression levels of 10 tomato ABF/AREB genes have a similar trend: the expression levels all reach the highest at 24 h (Fig. 10D). The biggest changes are seen in the expression levels of SlABF1, SlABF3, and SlABF5, which are increased by (14.98, 110.16 and 11.13 times compared with 0 h, respectively). The expression levels of SlABF1 and SlABF4 decrease at first and then increase with the extension of treatment time.

Discussion

ABA is an important plant hormone. Members of the ABF/AREB family are key transcription factors for ABA-dependent genes and they play important roles in plant hormone and abiotic stress responses (Chang et al., 2019; Kudoyarova et al., 2011). However, the ABF/AREB gene family in tomato has not been studied in detail. In this study, ten ABF/AREB genes were identified in the tomato genome. The 10 genes of ABF/AREB gene family were distributed on five chromosomes of tomato (Fig. 1). There are two pairs of tandem repeats and one pair of fragment repeats in the tomato ABF/AREB family. While seven ABF/AREB members were identified on five chromosomes in Solanum tuberosum (Liu et al., 2019), 10 ABF/AREB members were identified on eight chromosomes in Oryza sativa (Lu et al., 2009), and 14 ABF/AREB members on nine chromosomes in Populus trichocarpa (Ji et al., 2013). It can be seen that the more the number of chromosomes, the more the corresponding number of ABF/AREB genes. The possible reasons for its occurrence are the whole genome duplication event (WGD) and tandem gene duplication (Yong et al., 2021). In Actinidia chinensis, AchnABF2 is localized in the nucleus (Wei et al., 2020). Arabidopsis thaliana AREB/ABFs have been reported to localize in the nucleus and form heterodimers (Yoshida et al., 2010). In this study, the ABF/AREB family in tomato is mainly expressed in the nucleus, so another way to regulate the activity of tomato ABF/AREB proteins is dimerization. The secondary structures of AREB1 protein in Vignaum bellata and Phaseolus vulgari are alpha helix and random coil (Hai-long et al., 2018). Similarly, the secondary structure of ABF/AREB family in tomato in this study are mainly alpha helix and random coil. Therefore, irregular coil accounts for more, which may be because irregular coil connects more secondary structural elements.

ABF/AREBs structurally has five conserved domains (Fig. 2), 3 N-termini (C1, C2 and C3), 1 C-terminal DNA-binding bZIP region, and 1 terminal C4 conserved domain (Hong et al., 2013; Kim, 2005). ABA-dependent AREB1 is involved in gene regulation through multisite phosphorylation. For example, the phosphorylated active form of AREB1 can induce ectopic genes in vegetative tissues (Bastias et al., 2011). However, the hypothesis of the ABF/AREB phosphate site in tomato has not yet been established. This study shows that all the tomato ABF/AREB genes have a unique BRLZ domain at the C-terminus of the bZIP region, which functions to recognize and bind specific DNA sequences. The results indicate that ABF/AREB proteins are highly conserved in plant evolution. We analyzed the genetic structure of the tomato ABF/AREB and found that the intron numbers of the tomato ABF/AREB ranged from one to four. This is similar to the results of genetic structure of potato and Populus trichocarpa (Ji et al., 2013; Liu et al., 2019) and proving that the genetic makeup of the ABF/AREB family members is conserved (Fig. 5).

ABA activation is required in Arabidopsis AREB1/ABF2, AREB2/ABF4, and ABF3 to regulate ABRE-dependent signaling involved in drought stress tolerance (Yoshida et al., 2010). In this study, SlABF3, SlABF5, SlABF10 and AREB1/ABF2, AREB2/ABF4, ABF3 belong to the same grouping of the evolutionary tree (Group A). SlABF3, SlABF5 and SlABF10 are significantly induced by ABA, NaCl, UV, cold and PEG treatments. In addition, the three tomato ABF/AREB transcription factors showed similar expression patterns in cellular localization, genetic structure, and tissues. The results suggest that SlABF3, SlABF5 and SlABF10 play a redundant role in ABRE-dependent ABA signaling pathway under osmotic stress. However, another member of this subgroup, SlABF7 is expressed at a lower level than SlABF3, SlABF5 and SlABF10. The expression level of ABF1 in Arabidopsis is lower compared to AREB1/ABF2, AREB2/ABF4 and ABF3, but ABF1 is a functional homologue of AREB1/ABF2, AREB2/ABF4, and ABF3 dependent gene expression. The cellular localization of SlABF7 in genetic structure is similar to that of SlABF3, SlABF5 and SlABF10. Therefore, SlABF7 may be a functional homologue of SlABF3, SlABF5 and SlABF10.

It has been observed that promoter homeopathic elements are crucial for controlling gene expression, notably when biotic and abiotic stressors are present (Gao et al., 2021). We determined that the promoter region of the tomato ABF/AREB gene has a range of cis-acting components related to hormone response and abiotic stress (Fig. 7). This demonstrated that the ABF/AREB gene may be crucial for adapting to abiotic stressors and hormonal stimulation in tomato. Our functional verification research of the ABF/AREB gene revealed that ABA and PEG can activate several ABF/AREB genes (Vysotskii et al., 2013; Zandkarimi et al., 2015). Some ABF/AREB genes that are hormone-induced have corresponding hormone-related cis-elements in their promoters. For example, the relative expression levels of SlABF1, SlABF2, SlABF8 and SlABF10 are upregulated under osmotic treatment (Fig. 10), which is consistent with the distribution of osmotic response elements (MBS) in SlABF1, SlABF2, SlABF8 and SlABF10, implying that they might control gene transcription by combining active transcription factors with cis-acting components to produce the desired effects. Interestingly, there are also conflicting results in our analysis. SlABF3 and SlABF5 do not participate in the cryoresponsive element and responded significantly to osmotic stress. It might be because the transcription of the regulated genes is not influenced directly by the presence or absence of the appropriate cis-acting elements. In addition to the well-known ABA-induced phosphorylation by SnRK2 protein kinases, it has been demonstrated that Arabidopsis ABFs themselves are implicated in the induction of exogenous ABA treatment (Wang et al., 2019). This adds another layer of ABA control towards ABF proteins. It was found through the analysis of tomato ABF/AREB homeopathic elements that SlABF1-SlABF6, SlABF9 and SlABF10 possessed ABRE in their promoter regions (Fig. 7), implying that the rapid induction of their expression on exogenous ABA treatment might also be mediated by themselves.

ABF/AREB transcription factors participate in not only stress response, but also in hormone response. The role of ABF/AREBs in stress response, growth and development has been extensively studied and characterized in Arabidopsis thaliana and Solanum tuberosum (Li et al., 2013; Liu et al., 2019; Vishwakarma et al., 2017). ABF/AREB can bind to ABRE and activate the expression of ABA-dependent genes under drought stress (Fujita et al., 2011). It has been shown that the ABF/AREB family is sensitive to ABA response (Liu et al., 2019; Lu et al., 2009; Zandkarimi et al., 2015). The overexpression of TaAREB3 in Arabidopsis improved osmotic and freezing tolerance and enhanced ABA sensitivity (Wang et al., 2016). StCDPK2, a calcium-dependent protein kinase that phosphorylates StABF1 in vitro, is found to respond to ABA and NaCl (Muniz Garcia et al., 2012). However, we found that SlABF3 is significantly induced by ABA treatment. FLD is known as an inhibitor of ABA biosynthesis and FLD affects plant growth and development and stress response by reducing ABA levels (Ondzighi-Assoume, Chakraborty & Harris, 2016; Zou et al., 2018). The expression level of AchABF1-1 is induced by ABA but inhibited by FLD in Actinidia chinensis (Wei et al., 2022). In the present study, exogenous ABA promoted the expression of ABF/AREB in tomato. In contrast, FLD inhibited this promotion effect. These findings implied that the ABF/AREB genes are crucial for response of ABA in the tomato. We also discovered that tomato ABF/AREB genes respond significantly to drought stress. For instance, during osmotic stress, SlABF3 was considerably induced up to 110.16-folds, and SlABF5 was significantly induced up to 11.13-folds. It has been demonstrated that the ABF/AREB genes are crucial for response of osmotic stress in the tomato. This concurs with earlier research on Oryza sativa and Nicotiana tabacum (Hossain et al., 2010; Maruyama et al., 2012). Additionally, research revealed that excessive salt and osmotic pressure may also activate the majority of the ABA-induced genes (Seki et al., 2003). This suggested that there is an interaction between plant responses to hormones and abiotic stresses. Therefore the specific functions of tomato ABF/AREB genes need to be further investigated in depth.

Conclusions

In this study, a total of 10 tomato ABF/AREB gene family members were identified, which contains a common conserved structural domain and can be divided into two subfamilies. The gene family members are mainly expressed in the nucleus, and their secondary structures are mainly alpha helix and random coil. The gene family members contain several cis-acting elements associated with plant stress. SlABF2 and SlABF10 play an active role in fruit ripening. ABF/AREB can also respond to ABA and stresses. Additionally, SlABF3 is more sensitive under ABA treatment. SlABF1, SlABF3 and SlABF10 signifcantly respond to UV treatment. Osmotic stress greatly increases the expression of SlABF3 and SlABF5. Salt and cold stresses significantly induce SlABF3, SlABF5 and SlABF10. This study proposes a potential role for ABF/AREB in growth and abiotic stress response, and provides valuable candidate genes for improving stress resistance in tomato.

Supplemental Information

Supplemental Information 1 The ABFAREB protein sequences of Arabidopsis and nine Rosaceae species.

Click here for additional data file.

Supplemental Information 2 qRT-PCR data.

Click here for additional data file.

Supplemental Information 3 The Ka/Ks values of ABF/AREB genes in tomato.

Click here for additional data file.

Supplemental Information 4 Logos of the conserved motifs of ABFAREBs sequences.

Click here for additional data file.

Supplemental Information 5 The secondary structure elements distribution of ABFAREB gene family in tomato.

Click here for additional data file.

Supplemental Information 6 Cis-acting elements of SlABFs in promoter region.

Click here for additional data file.

Additional Information and Declarations

Competing Interests

Author Contributions

Data Availability

The authors declare that they have no competing interests.

Xuejuan Pan conceived and designed the experiments, performed the experiments, analyzed the data, prepared figures and/or tables, authored or reviewed drafts of the article, and approved the final draft.

Chunlei Wang conceived and designed the experiments, prepared figures and/or tables, authored or reviewed drafts of the article, and approved the final draft.

Zesheng Liu conceived and designed the experiments, performed the experiments, prepared figures and/or tables, and approved the final draft.

Rong Gao conceived and designed the experiments, analyzed the data, prepared figures and/or tables, and approved the final draft.

Li Feng conceived and designed the experiments, prepared figures and/or tables, analyzed the data, prepared figures and tables, and approved the final draft.

Ailing Li conceived and designed the experiments, prepared figures and/or tables, and approved the final draft.

Kangding Yao conceived and designed the experiments, prepared figures and/or tables, and approved the final draft.

Weibiao Liao conceived and designed the experiments, prepared figures and/or tables, and approved the final draft.

The following information was supplied regarding data availability:

The original data including tomato, potato, Arabidopsis and poplar ABF/AREB protein sequences are available in the Supplemental Files.

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
