# Peer review of "Identification of ABF/AREB gene family in tomato (Solanum lycopersicum L.) and functional analysis of ABF/AREB in response to ABA and abiotic stresses"

_PeerJ, doi:10.7717/peerj.15310_

## Round 0.1 · original submission · Major Revisions

Revise the manuscript as per the comments of the reviewers and resubmit for consideration.

Reviewer 1 ·

Basic reporting

Good experiment was conducted with proper replications and advance genomic softwares

Experimental design

Designing of experiment was appropriate

Validity of the findings

Finding are good and will be useful to further advancing of knowledge in this area. Few queries are raised which need to be satisfied

Additional comments

For comments see attached pdf

Annotated reviews are not available for download in order to protect the identity of reviewers who chose to remain anonymous.

·

Basic reporting

No comment

Experimental design

No comment

Validity of the findings

No comment

Additional comments

Reviewer comments:
Dear Editor thank you very much for giving me this opportunity of reviewing the manuscript entitled “Identification of ABF/AREB gene family in tomato (Solanum lycopersicum) and functional analysis of ABF/AREB in response to ABA and abiotic stresses”. The research idea was very well designed and also written well, but it is not ready for publication in its present format. The decision over the manuscript is “Major Revision”. All the required corrections are highlighted inside the manuscript with attached comment boxes. Authors are asked to go through all of them and correct them.
Comments:
Title: In addition to the scientific name write the identifier name as well.
Abstract:
Line number 16-19:
Start your abstract with a rational sentence depicting the importance of the study. After that author can write the aims and objective followed by standard methodologies or techniques used in the study.
Line number 19-20:
This particular statement needs to be presented in more of a understandable way in the revised version of your manuscript.
Line number 32-34:
That is okay but what will be the future prospective or the importance of the study in the present-day scenario will be. Need to write one or two sentences regarding above mentioned statement as concluding remark.
Keywords:
Arrange all the keywords in alphabetical order.
Introduction:
Line number 45-47:
These are two different sentences or a single sentence. Because there is one full stop "." between the word’s development and drought why??
Avoid such kind of mistakes in the revised version of the manuscript.
Line number 49:
Write one small paragraph about the species of concern also.
Line number 65-73:
The gaps or the need of the study are well established but the author failed to write a proper introduction for the study. Author needs to link paragraph to paragraph. For example, here author must not Jump from one topic to another randomly. The introduction section needs to be in a proper flow.
Line number 79-80:
Novelty of the study is missing author need establish some points where novelty of the current study can be better understood.
Materials and methods:
Line number 84-85:
Write the original source name.
Line number 87:
Write scientific names in italics.
Line number 95:
Author can use the website from where it was purchased or subscribed.
Line number 136-137:
This is okay but for all the above-mentioned methods, is there any earlier reference available?? If yes then cite some valid references.
Line number 142-143:
Provide the altitude, latitude and longitude of the place.
Line number 152, 153, 155, 158:
Express this mM's in M only. There is no space required while writing digit and percentage symbol. i.e. 20 % should be written as 20%. Follow the pattern throughout the whole manuscript. Uv should be written as UV. Which was the instrument used for storing the samples at -80 Degree Celsius??
Need to mention the instrument name with company within bracket.
Line number 159:
The font size is not in accordance to the Journal format for the wordings:
"RNA extraction and quantitative qRT-PCR"
Results, Discussion:
Line number 175: 10 tomato: Write as Ten tomato...
Line number 311-313, 314: For this particular statement author need cite some valid references. Avoid the use of words such as we, our, us etc.
Line number 320-321: Is there any other probable explanation there. If yes, then mention them with appropriate references.
Line number 320-321: Is there any other probable explanation there. If yes, then mention them with appropriate references.
Conclusions: Line number 415-416: Avoid the use of word "our". Write the novelty, importance of the study in the present-day scenario and future prospective of the study.
References: All the references strictly must be in accordance to the Journal format.
Author can go through the authors guideline section for further assistance.
Additionally, the language of the manuscript must be revised for any grammatical or typical errors.

·

Basic reporting

In the manuscript authors have identified ABF/AREB gene family members in tomato (Solanum lycopersicum) and further performed functional analysis of ABF/AREB members in response to ABA and abiotic stresses. Some concerns related to manuscripts are:
1. In the introduction authors should include latest statistics about tomato production, productivity from FAO etc. Also authors should replace the old references used in the introduction with the maximum 5 years old articles.
2. Avoid using loose sentences, for example, line 70, “Therefore, exploring the potential functions of genes ……to stress”. Which genes are you focusing on, and why are they essential? Please mention it here.
3. What do you mean by homeopathic promoter analysis (Line 76)?
4. The manuscript is very weak regarding spelling errors, typos, English grammar, and other disturbing minor mistakes. Consistency of terms, expressions (for example, E value of 1E-20), and abbreviations used are not maintained in the manuscript. Authors should take the help of native speakers of English for manuscript reading and correction.

Experimental design

1. Give a reference to TB tools used in the study.
2. Name the online tool used for secondary protein structure determination. Also, in the materials and method section, please mention all the software and programs used for each bioinformatics analysis.
3. What is 1/2 nutrient solution mentioned in the plant material section?
4. Give the reference for the 2-&&Ct method used in the study.

Validity of the findings

1. You have discussed the tandem and whole genome duplication event in the discussion for number variation between species. Have you checked these events in your study? If not, please include it in the revised version.
2. In table 5, what do you mean by homeopathy?
3. Conclusion should be rewritten to emphasize the novelty of work and future research that can be done based upon your research.

---

## Round 0.2 · accepted · Accept

All the comments have been resolved properly now the manuscript is ready for publication

Reviewer 1 ·

Basic reporting

I reviewed the original article few months back. Language has substantially improved and introduction has been revised drastically.

Experimental design

In my opinion author has used all well established tools to identify the AREB/ABF TF elements followed by its characterization and expression analysis.

Validity of the findings

Findings has been validated with real time PCR analysis suggested that identified TF works well in stress conditions.

Additional comments

Author can add few sentences how to exploit these TF for crop improvement in conclusion section.

·

Basic reporting

Reviewer comments:
The manuscript “#80477” entitled “Identification of ABF/AREB gene family in tomato ( Solanum lycopersicum L.) and functional analysis of ABF/AREB in response to ABA and abiotic stresses” has been well revised by the authors and addressed all the required corrections. The decision over the manuscript is “Accept.”

Experimental design

NA

Validity of the findings

NA

Additional comments

NA

·

Basic reporting

Authors have improved the MS significantly, and the MS is suitable to fit for publication.

Experimental design

Authors have improved the MS significantly, and the MS is suitable to fit for publication.

Validity of the findings

Authors have improved the MS significantly, and the MS is suitable to fit for publication.